# Coding of object location by heterogeneous neural populations with spatially dependent correlations in weakly electric fish

**Myriah Haggard**[1], **Maurice J. Chacron**[2]*

**1** Quantitative Life Sciences, McGill University, Montreal, Canada, **2** Department of Physiology, McGill University, Montreal, Canada

* maurice.chacron@mcgill.ca

**Data Availability Statement:** All data files and codes are available from the figshare database (https://doi.org/10.6084/m9.figshare.21318321).

## Abstract

Understanding how neural populations encode sensory stimuli remains a central problem in neuroscience. Here we performed multi-unit recordings from sensory neural populations in the electrosensory system of the weakly electric fish *Apteronotus leptorhynchus* in response to stimuli located at different positions along the rostro-caudal axis. Our results reveal that the spatial dependence of correlated activity along receptive fields can help mitigate the deleterious effects that these correlations would otherwise have if they were spatially independent. Moreover, using mathematical modeling, we show that experimentally observed heterogeneities in the receptive fields of neurons help optimize information transmission as to object location. Taken together, our results have important implications for understanding how sensory neurons whose receptive fields display antagonistic center-surround organization encode location. Important similarities between the electrosensory system and other sensory systems suggest that our results will be applicable elsewhere.

## Author summary

Despite decades of research, the functional roles of neural heterogeneities towards understanding how sensory inputs are encoded by neural populations remains poorly understood. Here we use multi-unit recordings from sensory neural populations using high-density arrays (i.e., Neuropixels probes) and mathematical modeling to understand how a heterogeneous neural population with antagonistic center-surround receptive field organization encodes object location. We recorded the activities of pyramidal cells within the electrosensory lateral line lobe of weakly electric fish in response to a prey-like stimulus. Overall, we found that the receptive fields were highly heterogeneous even when they overlap considerably. We also found that correlated trial-to-trial variabilities of neural responses (i.e., spike-count correlations) varied along the receptive field. Specifically, correlation magnitude was highest towards the receptive field edges and dropped sharply near the midpoint. Using Fisher information analysis, we determined that the spike-count correlations introduced redundancy, but that the deleterious effect was in part mitigated by their spatial dependence. To better understand how heterogeneities within the

**Funding:** Canadian Institutes of Health Research http://www.cihr-irsc.gc.ca/ (grant number 159694); received by MJC. The funder had no role in study design, data collection and analysis, decision to publish, or preparation of the manuscript.

**Competing interests:** The authors have declared that no competing interests exist.

receptive field, as well as spatially dependent correlations, influence information transmission, we built a mathematical model. Overall, our model reproduced experimental data and predicted that the level of heterogeneity in receptive field position seen experimentally is optimal for information transmission. Given that there are important parallels between the electrosensory system and other senses (e.g., vision), it is likely that our results will be applicable elsewhere.

## Introduction

Understanding how the activities of neural populations are combined in the brain remains an important area of research in neuroscience. Indeed, population coding has been studied extensively, both experimentally and theoretically [1–11]. Yet, despite decades of work, how populations of neurons represent sensory input remains poorly understood [2]. This is, on the one hand, due to the fact that neurons display large heterogeneities in their responses to sensory input [12–14], which can under some circumstances benefit information transmission by increasing the coding range [1, 15–24]. On the other hand, correlations between neural activities in the form of signal correlations (i.e., correlations between the average responses of neurons to stimuli) as well as spike-count correlations, also known as noise correlations (i.e., correlations between the trial-to-trial variabilities of neurons), can strongly influence information transmission by neural populations [3, 6, 10, 11, 25]. Further complexity arises because correlations are highly plastic and are regulated by attention [26, 27], single neuron firing properties [28], and differential stimulus features corresponding to differential behavioral contexts [29–31] (see [32] for review). Here we investigated how sensory neural populations encode information about prey location, which is crucial for informing behavior thereby ensuring survival.

Gymnotiform wave-type weakly electric fish are a tractable model system for understanding the effects of neural heterogeneities and correlations on population coding due to their well-characterized anatomy [33] and physiology [34–37]. These fish generate a quasi-sinusoidal electric field around the body through the electric organ discharge (EOD) and can sense amplitude modulations of this field through an array of electroreceptors on the skin surface that synapse onto pyramidal cells within the electrosensory lateral line lobe (ELL) [38]. ELL pyramidal cells have antagonistic center-surround receptive field organization [39, 40] and display large heterogeneities in terms of cell morphology, distribution of ion channels, as well as firing properties and responses to sensory input (see [41] for review). Behavioral studies have shown that these fish can effectively detect and capture prey even though the electric signals that prey generate are very faint and only impinge on a small fraction of the sensory epithelium [42–45]. Previous studies have looked at the neural mechanisms mediating both prey detection [46–49] and estimation of location [50, 51]. However, these studies were either conducted using single unit recordings or used models that did not include the full characteristics of ELL pyramidal cell receptive fields in that only the center portion was considered. Thus, the effects of the full receptive field structure as well as how correlations between the activities of ELL pyramidal cells influence their ability to transmit information as to prey location remains unknown to date.

Here we used high-density electrode arrays to record the activities of ELL pyramidal cells simultaneously in response to local stimuli located at different positions along the fish's rostro-caudal axis. Overall, we found that ELL pyramidal cells displayed heterogeneous receptive fields and that spike-count correlations between their activities were highly spatially

dependent. Specifically, correlation magnitude was highest at the edges and lowest at the middle of the region of receptive field overlap. Computing Fisher information revealed that sufficient information as to stimulus location was contained in the activities of populations of ~19 neurons. While spike-count correlations introduced redundancy and lowered information transmission, their spatial dependence helped offset such redundancy as compared to spatially independent correlations. A mathematical model that included key aspects such as observed heterogeneities in receptive fields and spatially dependent correlations reproduced experimental results. Varying the level of receptive field heterogeneity in the model revealed that information is optimized for a given level that is similar to that observed experimentally. Taken together, our results show that several mechanisms such as receptive field heterogeneity and the spatial dependence of correlations help increase the information transmitted by ELL pyramidal cell populations about object position. Because of important similarities between the electrosensory system and other systems (e.g., visual) [52], it is likely that our results will be applicable elsewhere.

## Results

Here we investigated how spike-count correlations and neural heterogeneities impact the coding of spatial position by ELL pyramidal cell populations. To mimic the electric image caused by a prey stimulus, we delivered a 4 Hz sinusoidal amplitude modulation of the animal's own EOD through a small dipole located near the skin. The dipole was moved along the animal's rostro-caudal axis (Fig 1, upper right), only stimulating a small portion of the sensory

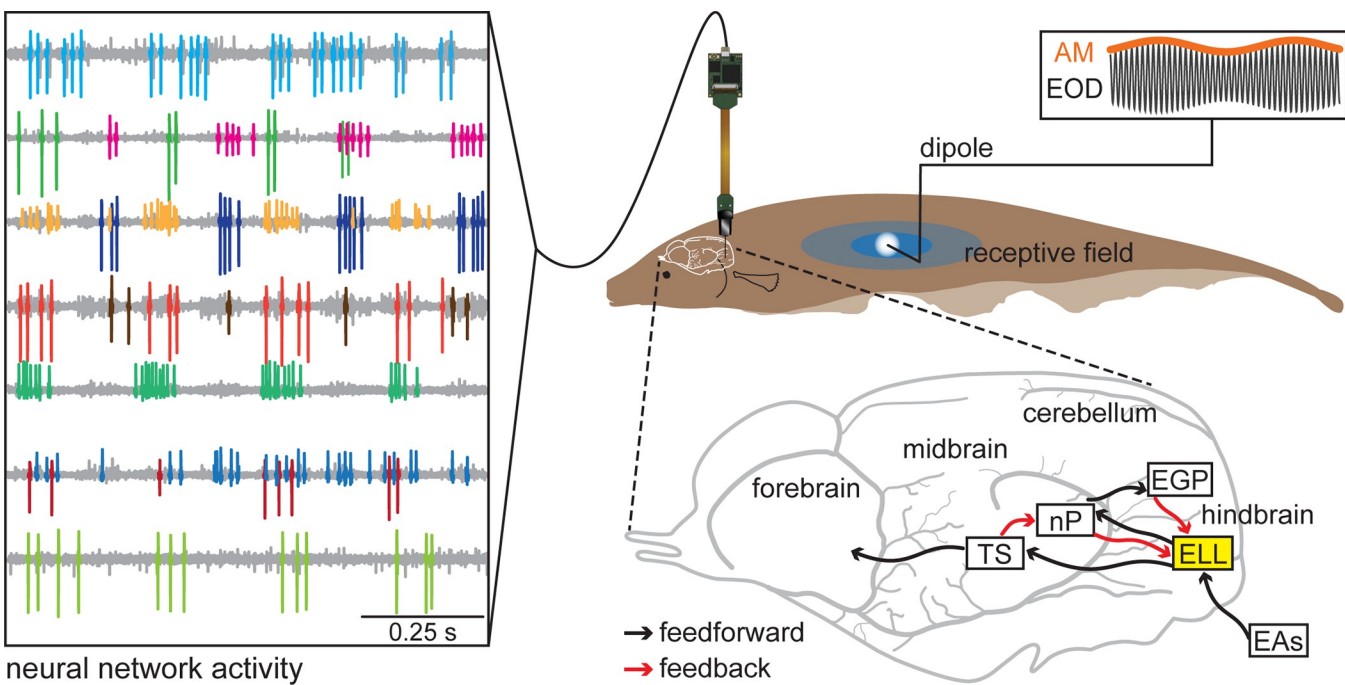

**Fig 1. Experimental setup.** The simultaneous activity of a population of neurons is recorded with a Neuropixels probe while the fish is stimulated locally with a 4 Hz prey-mimic amplitude modulation (upper right: EOD black, AM orange). Example raw voltage traces, with the spiking activity of each neuron labeled in a different color, is shown on the left. The dipole delivering the stimulus is placed at positions along the length of the fish within the receptive fields of the neurons being recorded (schematic receptive field on fish's skin: center in dark blue, surround in lighter blue). The electric image of the stimulus delivered through the dipole electrode projects onto the surface of the fish's skin diffusely (white circle). The electrosensory circuitry of the brain (bottom right) is composed of feedforward (black arrows) and feedback pathways (red arrows), with the ELL highlighted in yellow. EAs, electrosensory afferents; ELL, electrosensory lateral line lobe; TS, torus semicircularis; nP, nucleus praeeminentialis; EGP, eminentia granularis posterior.

epithelium per stimulus position as done previously [39]. Simultaneous recordings from multiple ELL pyramidal cells were achieved by a high-density electrode array (Neuropixels probe; Fig 1, left) that was oriented to record from ELL pyramidal cells whose receptive field centers were tuned to approximately the same position on the fish's rostro-caudal axis (see Materials and Methods). The dipole was then moved at 0.5 cm intervals along the fish's rostro-caudal axis and the stimulus repeated, which allowed us to study how much information about spatial location is carried by the spiking activities of a population of ELL pyramidal cells. As mentioned above, ELL pyramidal cells receive input from electroreceptor afferents and project to higher brain areas that mediate perception and behavior (Fig 1, lower right; black arrows). Additionally, ELL pyramidal cells receive large amounts of descending input from higher brain areas [53] (Fig 1, lower right; red arrows).

## ON and OFF-type ELL pyramidal cells display antagonistic center-surround receptive fields

We first mapped the receptive fields of the recorded neurons as a function of spatial position along the rostro-caudal axis (Fig 2A). Examples of spiking activity obtained during 4 consecutive stimulus trials (orange) at three different spatial positions are shown for typical ON- and OFF-type neurons (Fig 2B and 2C: left, middle and right; green and purple curves, respectively). For the example ON-type neuron, spiking occurred primarily during the positive half-cycle of the sinusoidal stimulus within the receptive field center, but occurred instead during the negative half-cycle within the surround (Fig 2B). In contrast, for the example OFF-type neuron, spiking occurred primarily during the negative half-cycle of the sinusoidal stimulus within the receptive field center and during the positive half-cycle within the surround (Fig 2C). Note that we only considered responses during the positive half-cycle of the stimulus, as these mimic the increase in EOD amplitude caused by prey stimuli [42]. As such, receptive fields of ON and OFF-type neurons displayed opposite shapes (compare green curve in Fig 2B to purple curve in Fig 2C). Fig 3 shows the receptive fields of 32 neurons recorded from simultaneously, grouped into ON-type (top) and OFF-type (bottom). While receptive fields were located at more or less the same position and therefore largely overlapped, they still displayed large heterogeneities even for cells of a given type (e.g., ON-type cells).

## ELL pyramidal cells display spatially dependent spike-count correlations

Knowing the shape of the receptive field is not sufficient to quantify information transmission by neural populations. This is because information also depends on correlations between neural responses as mentioned above. As such, we computed spike-count covariance matrices between neural responses for each stimulus position (see Materials and Methods). Overall, we found that covariance varied as a function of stimulus position (i.e., was spatially dependent) in a consistent manner across recording sessions. Fig 4 shows three example covariance matrices obtained for three different spatial positions. When the stimulus was delivered in the middle of the receptive field overlap, where there is primarily overlap between the receptive field centers, we found weak covariation between neural responses (Fig 4, center). In contrast, when the stimulus was delivered towards the edges of the receptive field overlap, where there is primarily overlap between the receptive field surrounds, the covariation was larger in magnitude (Fig 4, left and right panels). As such, covariance magnitude was significantly lower when stimulating near the overlap middle than at the overlap edges (Fig 4, inset of center panel; Kruskal-Wallis: $p = 1.4 \cdot 10^{-52}$; left-center: $p = 3.5 \cdot 10^{-51}$; left-right $p = 4.8 \cdot 10^{-6}$; center-right $p = 1.7 \cdot 10^{-24}$).

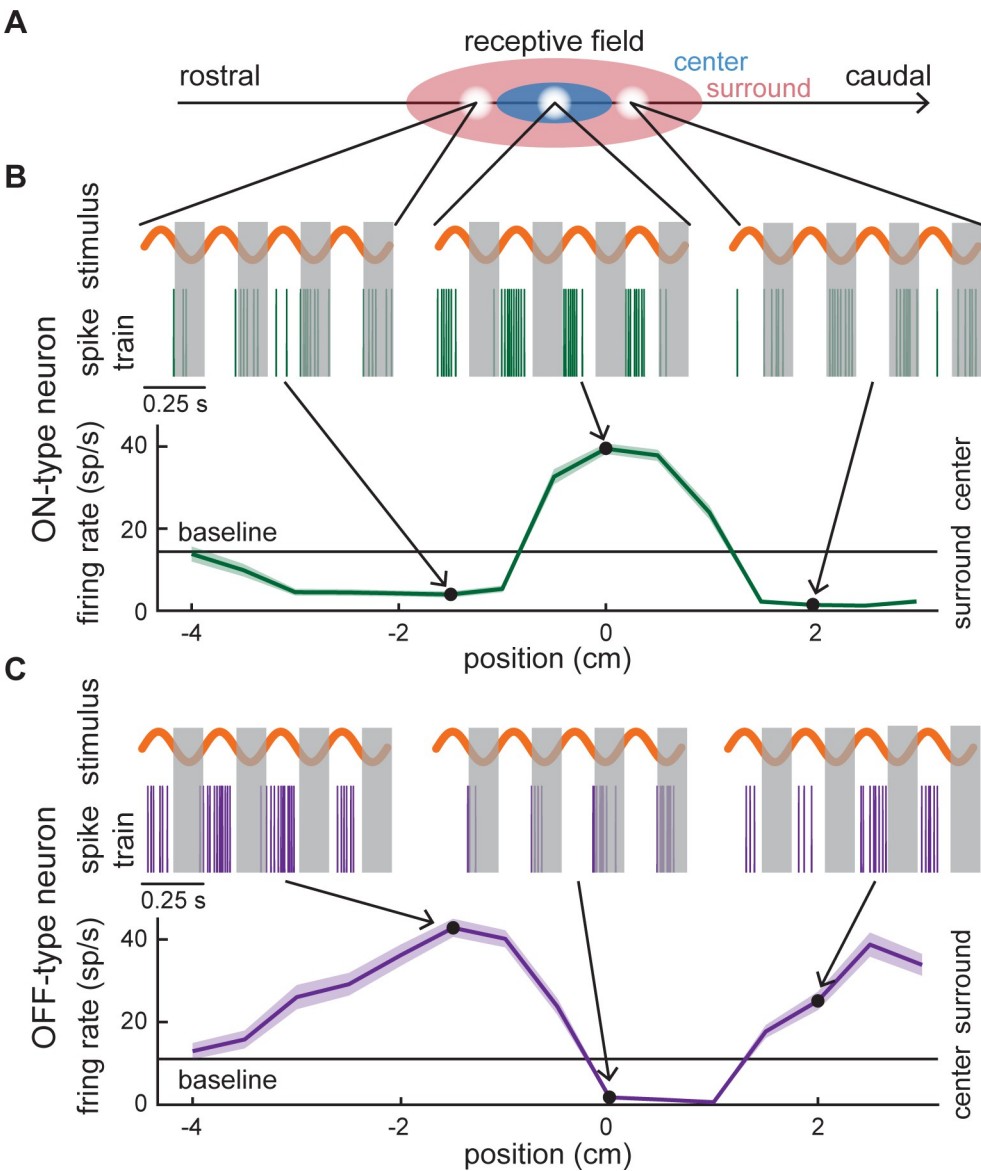

**Fig 2. Mapping the receptive fields of ON and OFF-type pyramidal cells.** (A) Schematic of a receptive field with the center (blue)—surround (red) organization and stimulation paradigm. A dipole delivering a local stimulus (white circles) is systematically moved from rostral to caudal positions to map the receptive fields of the recorded neurons. (B) *Top*: Neural responses from an example ON-type pyramidal cell to stimulation at different locations on the rostro-caudal axis (spike train, green). The positive half-cycle of each repetition, or trial, of the stimulus (orange) was analysed. The grey bands cover the negative half-cycle of the stimulus. *Bottom*: Trial-averaged firing rate (green) as a function of stimulus position (i.e., the receptive field) for this example cell, with shaded error bars indicating the SEM. The responses corresponding to the three positions shown in (A) are marked by arrows and black dots. The receptive field center and surround were defined as the regions in space for which the firing rate was either greater or lesser than the baseline firing rate (horizontal black line), respectively. (C) Same as (B), but for an example OFF-type pyramidal cell (purple). In this case the receptive field center and surround were defined as the regions in space for which the firing rate was either lesser or greater than the baseline firing rate, respectively.

Why are covariances between neural responses spatially dependent? In theory, changes in covariance could be due to either changes in correlations between neural responses or changes in the variances of the responses themselves, as the covariance is the product of the correlation coefficient and the variance (see Materials and Methods). Thus, we computed correlations

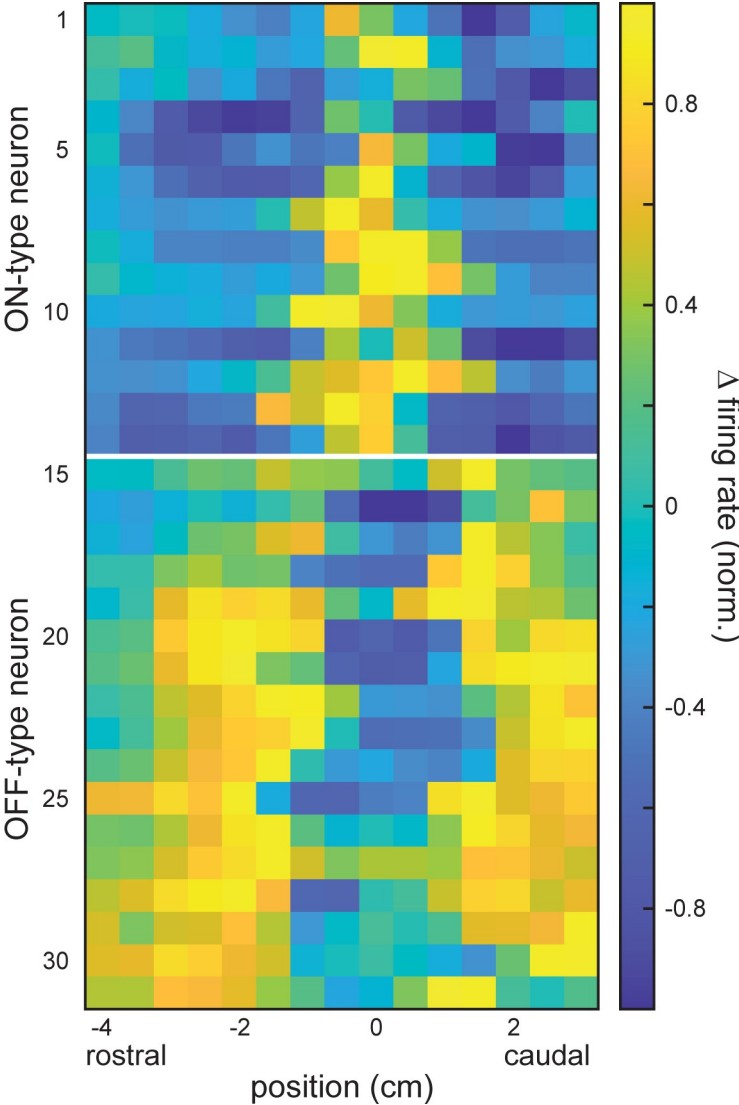

**Fig 3. Example receptive fields from ON and OFF-type ELL pyramidal cell populations.** Color plot showing the receptive fields of 32 simultaneously recorded neurons (14 ON-type neurons listed first, followed by 17 OFF-type neurons) visualized as the change (Δ) in normalized firing rate as a function of stimulus position.

between neural responses (i.e., spike-count correlations; S1 Fig). Further, because previous studies have shown that correlations between neural responses are strongly influenced by correlations between the baseline activities (i.e., in the absence of stimulation) [29, 54], we investigated whether, and if so, how the spatial dependence of correlations under stimulation varies as a function of baseline correlations. Overall, we found that baseline correlations varied over a wide range and decayed in magnitude as a function of the distance between neurons (Fig 5A; see S2 Fig for the raw baseline correlation values), which agrees with previous results [29, 54]. By applying a threshold at $|r_{BL}| = 0.15$ (Fig 5A, red line), we separated neural pairs into those with high baseline correlation magnitude (i.e., "high $|r_{BL}|$ pairs") and those with low baseline correlation magnitude (i.e., "low $|r_{BL}|$ pairs"). We then found that the spatial dependence of the correlation was greater for high $|r_{BL}|$ pairs than for low $|r_{BL}|$ pairs (Fig 5B and 5C; Wilcoxon rank sum test for b: $p = 2.1 \cdot 10^{-128}$). Indeed, for high $|r_{BL}|$ pairs, spike-count

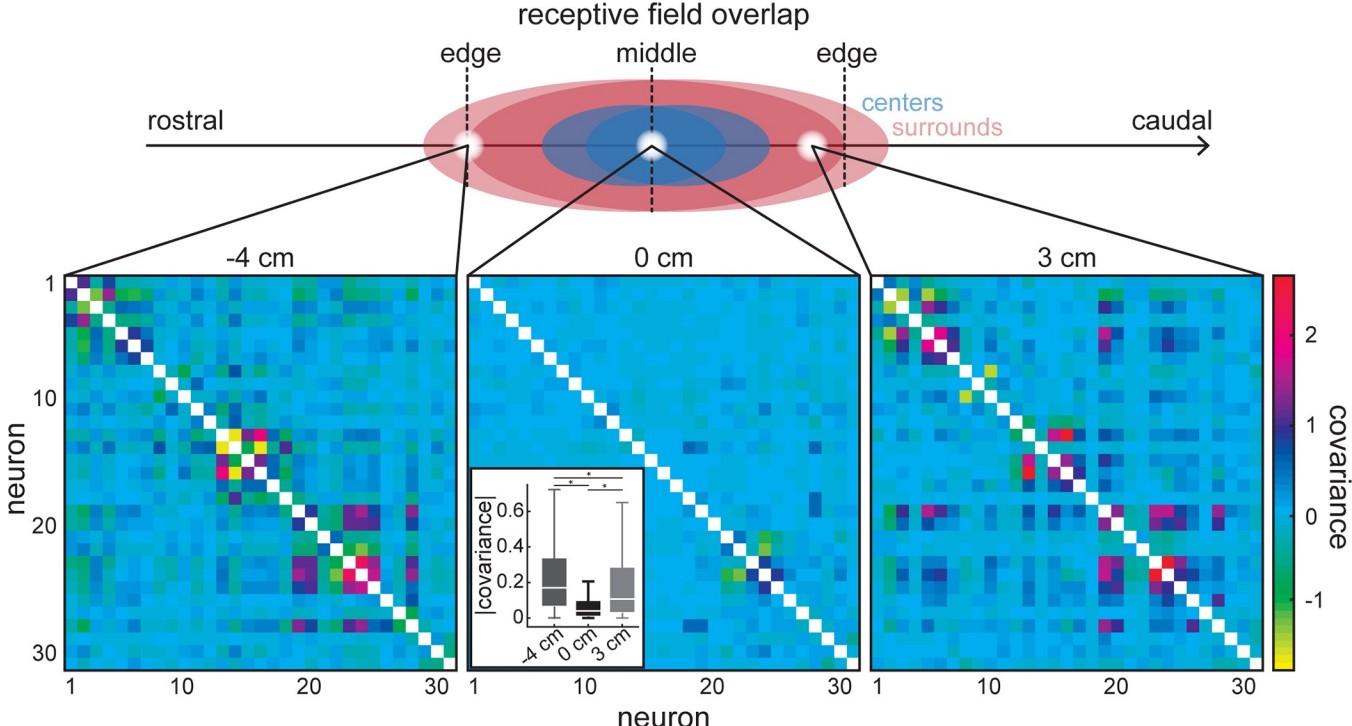

**Fig 4. ELL pyramidal cells display spatially dependent covariance.** Covariance matrices (bottom panels) obtained for three different positions (top; receptive field schematic for two overlapping neurons: centers in blue, surrounds in red). Pairwise covariances at stimulus positions near the edges of where the receptive fields overlap (left and right panels), where the surrounds of most neurons are activated, are larger than in the middle of the receptive field overlap (center panel), where the centers of most neurons are activated. (Note: the variances along the diagonal are not shown to emphasize the spatial dependence of the covariances.) Center inset: The distribution of covariance magnitudes is significantly different across the three stimulus positions (Kruskal-Wallis: p = 1.4·$10^{-52}$; left-center: p = 3.5 · $10^{-51}$; left-right p = 4.8 · $10^{-6}$; center-right p = 1.7 · $10^{-24}$). "*" indicates statistical significance.

correlations at the overlap edges approached the baseline value while those at the overlap middle were near zero, leading to a strong spatial dependence (Fig 5C, left). In contrast, for low |$r_{BL}$| pairs, spike-count correlations were near zero for all locations, leading to weak or no spatial dependence (Fig 5C, right). Finally, we quantified spike-count correlation magnitude in regions of center-center (i.e., overlap middle), center-surround, and surround-surround (i.e., overlap edges) overlap (blue, pink, and green, respectively) for all pairs across datasets. Overall, the correlation distribution for center-center overlap had a significantly lower median than that computed for center-surround which in turn was lower than that of surround-surround overlap when considering high |$r_{BL}$| pairs only (Fig 5D, left; Kruskal-Wallis: p = 1.8 · $10^{-131}$; cc-cs: p = 9.7 · $10^{-10}$; cc-ss: p = 9.6 · $10^{-10}$; cs-ss: p = 9.6 · $10^{-10}$). In contrast, when considering low |$r_{BL}$| pairs only, all three distributions overlapped (Fig 5D, right; Kruskal-Wallis: p = 0.89; cc-cs: p = 0.94; cc-ss: p = 0.89; cs-ss: p = 0.99). Thus, our results show that the spatial dependence of the covariance is due to the spatial dependence of spike-count correlations and that neural pairs with high baseline correlations display the greatest spatial dependence.

## Information transmission by ELL pyramidal cell populations

We computed the Fisher information in order to quantify the accuracy by which an optimal linear decoder can estimate the location of the stimulus given the recorded activities of ELL pyramidal cells. The Fisher information is the inverse variance of the optimal linear estimator and is calculated from the derivatives of the receptive fields as well as the covariance matrix for

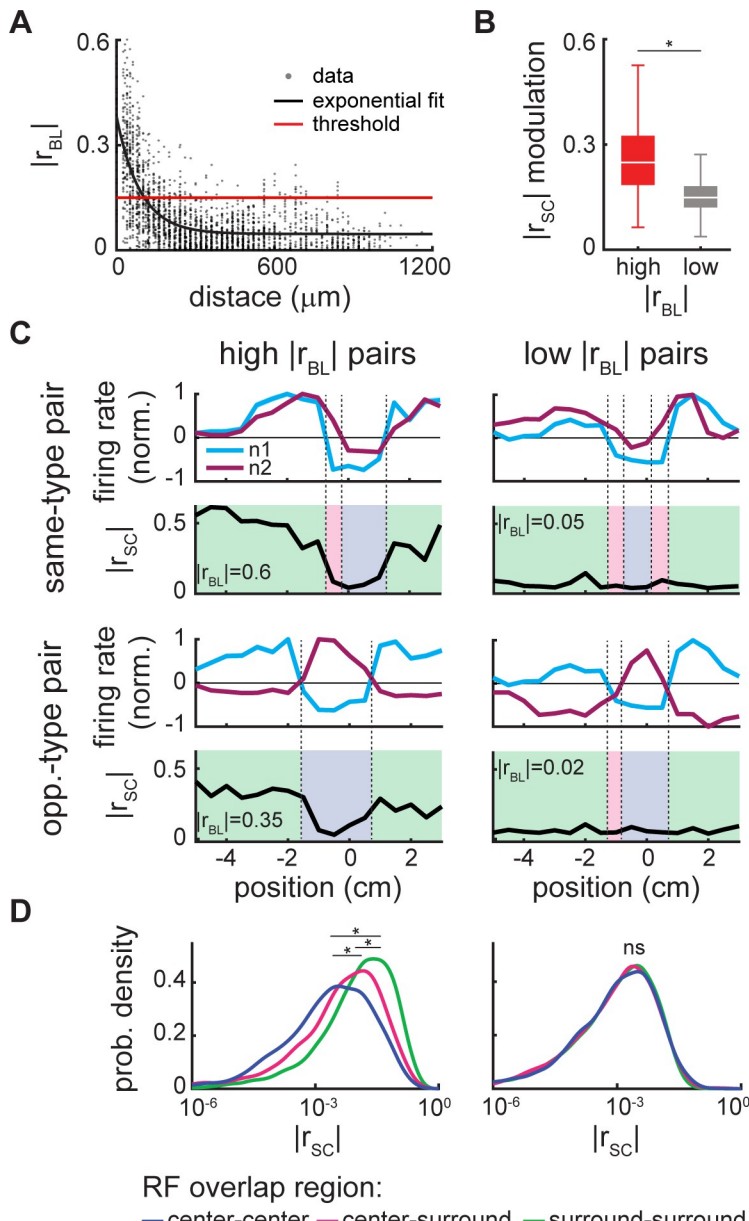

**Fig 5. Spatial modulation of spike-count correlations depends on the level of baseline correlations.** (A) Magnitude of pairwise baseline correlations ($|r_{BL}|$; black circles) as a function of the relative distance between neurons with a fitted exponential curve (black line). A threshold of $|r_{BL}| = 0.15$ (red line) was used to separate neural pairs into high and low $|r_{BL}|$ pairs. (B) The spatial depth of modulation (max.-min.) of the spike-count correlations ($|r_{SC}|$) is significantly higher for high $|r_{BL}|$ pairs than for low $|r_{BL}|$ pairs across recording sessions (Wilcoxon rank sum test for b: $p = 2.1 \cdot 10^{-128}$; outliers were removed as detailed in the Materials and Methods section). (C) Spatial dependence of the $|r_{SC}|$ is shown for four example pairs, two same-type pairs (OFF-OFF type pairs) and two opposite-type pairs (ON-OFF type pairs) each with low and high $|r_{BL}|$. For high $|r_{BL}|$ pairs (left column), the spatial dependence of the $|r_{SC}|$ (black) is determined by the region of receptive field overlap (receptive field neuron 1 blue, receptive field neuron 2 purple): in regions where the surrounds of both neurons overlap, the $|r_{SC}|$ approaches $|r_{BL}|$ (green background); in regions where the surround of one neuron overlaps the center of the other, the $|r_{SC}|$ is intermediate (pink background); and in regions where there is center-center overlap, the $|r_{SC}|$ approaches zero (blue background). In contrast, for low $|r_{BL}|$ pairs of either type (right column), there is minimal or no spatial modulation and the magnitude hovers near zero. (D) Distributions of $|r_{SC}|$ for regions of center-center (blue), center-surround (pink) and surround-surround overlap (green) pooled over all pairs across all recording sessions: for high $|r_{BL}|$ pairs (left) the three distributions are significantly different (Kruskal-Wallis: $p = 1.8 \cdot 10^{-131}$; cc-cs: $p = 9.7 \cdot 10^{-10}$; cc-ss: $p = 9.6 \cdot 10^{-10}$; cs-ss: $p = 9.6 \cdot 10^{-10}$); however, they are not significantly different for low $|r_{BL}|$ pairs (right; Kruskal-Wallis: $p = 0.89$; cc-cs: $p = 0.94$; cc-ss: $p = 0.89$; cs-ss: $p = 0.99$). "*" indicates statistical significance; ns, not significant; c, center; s, surround.

each spatial location (Fig 6A; see Materials and Methods). In order to investigate the role of spatially dependent covariance on information transmission, we considered three different scenarios shown in Fig 6B. In the first scenario, covariances were dependent on position as observed in the data (i.e., spatially dependent; Fig 6B, left). In the second scenario, we set all off-diagonal elements of the covariance matrix to zero, thereby effectively eliminating spike-count correlations (i.e., independent; Fig 6B, middle). In the third scenario, we rendered the spike-count correlations spatially independent by replacing the spatially dependent correlations for each neural pair with the baseline correlation of that pair (Fig 6B, right) and then recalculated the covariance matrices using these values. Our results show that the Fisher information was greatest around the middle of the population of receptive fields (Fig 6C) for all scenarios considered. However, while Fisher information values were lower with spatially dependent correlations than without correlations (Fig 6C, compare blue and red), Fisher information values obtained with spatially independent correlations were even lower (Fig 6C, compare blue and purple). Therefore, while spatially dependent spike-count correlations had an overall detrimental effect on information transmission through increased redundancy, the spatial dependence greatly attenuated this redundancy and was thus beneficial. The relationship between Fisher information values computed with spatially dependent correlations, without correlations (i.e., independent), and with spatially independent correlations were seen for all neural population sizes (Fig 6D). Indeed, in all cases the Fisher information increased linearly as a function of population size with values obtained for spatially dependent correlations below those obtained for no correlations and above those obtained for spatially independent correlations (Fig 6D inset; one-way ANOVA: df = 12, p = $3.8 \cdot 10^{-6}$; SD—I: p = $6 \cdot 10^{-4}$; SD—SI: p = $6.3 \cdot 10^{-3}$; I—SI: p = $2.8 \cdot 10^{-6}$).

To estimate the precision of an optimal linear decoder at determining the stimulus' location along the fish's rostro-caudal axis, we computed the square root of the Cramér-Rao bound and compared this value to the size of a typical prey. Overall, the square root of the Cramér-Rao bound was below the prey radius (Fig 6E), indicating that there is enough information to accurately estimate the location of the prey. A plot of the root Cramér-Rao bound as a function of population size revealed that a population of ~19 neurons is sufficient to accurately estimate prey location (Fig 6F).

## Modeling ELL population spiking activities

So far, we have shown that spatially dependent correlations can limit the detrimental effects that would be observed had they been spatially independent, and that information as to stimulus location is available from populations of ~19 neurons. In order to better understand how different characteristics of ELL pyramidal cell responses affect information transmission, we built a mathematical model that incorporated heterogeneities in receptive fields as well as spatially dependent correlations (Fig 7A, see Materials and Methods). Specifically, we modeled a population of neural receptive fields (N = 32, the mean population size across recording sessions), each built as a difference of Gaussians with the receptive field position, width and amplitude drawn from distributions fitted to the data (S3 Fig). We then fit the variance as a linear function of the firing rate and the baseline correlations as a function of relative distance with an exponential function (S4 Fig). Finally, we modeled the spatially dependent spike-count correlations as a function of receptive field center overlap and baseline correlations (S5 Fig). This generates spatially dependent covariance matrices similar to the data (compare S6 Fig with Fig 4). Overall, we found that this model could qualitatively reproduce the dependency of the Fisher information on position and population size with values obtained for spatially dependent correlations below those obtained for no correlations and above those

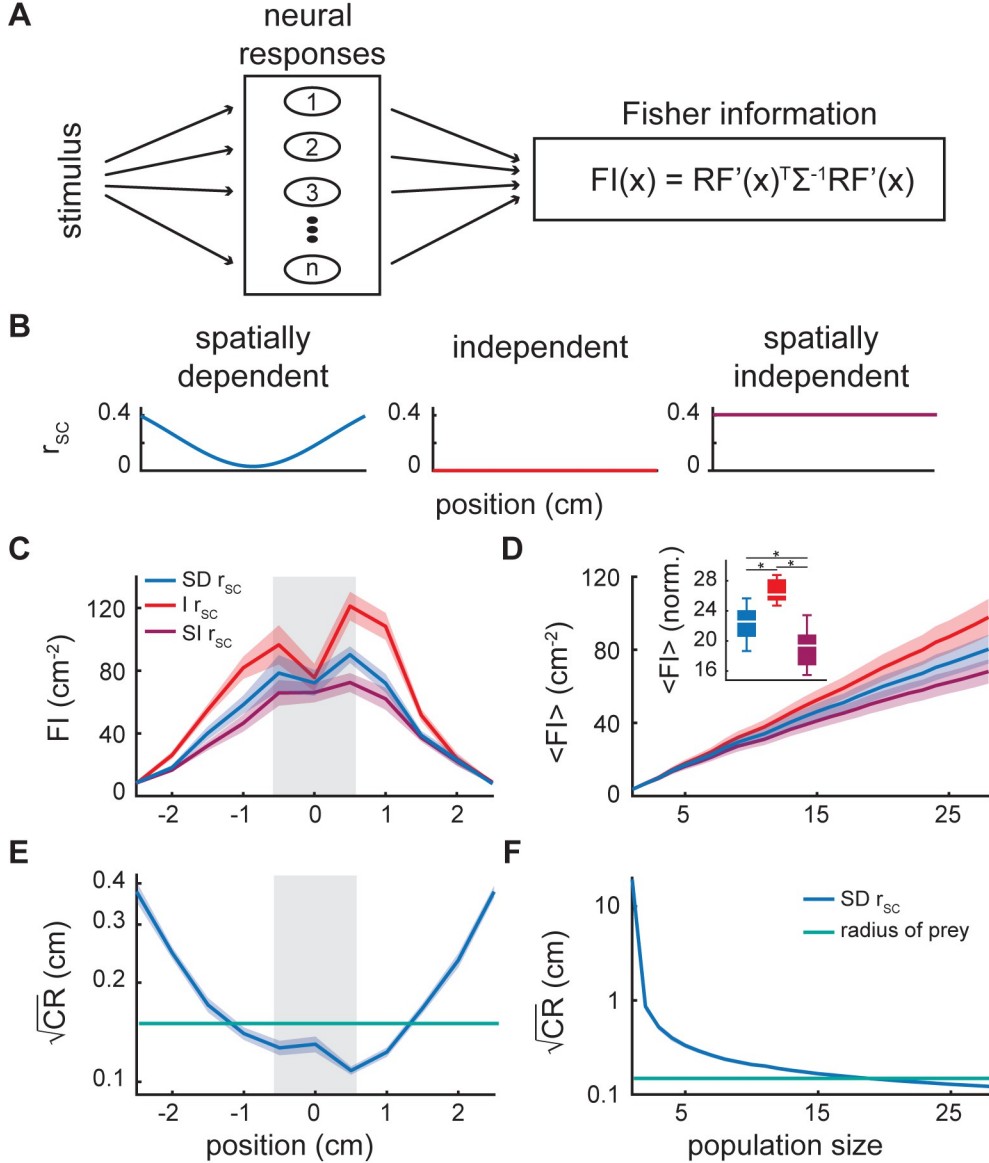

**Fig 6. Redundancy reduction by spatially dependent spike-count correlations.** (A) Schematic for calculating Fisher information (FI) from the neural responses (i.e., the derivative of the neural receptive fields and the covariance matrices). (B) Schematic of the different correlation scenarios used when calculating the Fisher information: the intact data which includes spatially dependent $r_{SC}$ (SD: left, blue), the independent case where all correlations are set to zero (I: middle, red) and the spatially independent case where the correlation values at all stimulus positions for a given pair are set to the baseline correlation for that pair (SI: right, purple). (C) Fisher information as a function of stimulus position for the three correlation scenarios, averaged over recording sessions with a population size of 28 neurons. The dip in Fisher information at position zero is caused by the slopes of many neurons in the population approaching zero as the firing rates reach their maximum. The light grey bar indicates the region of peak information and the stimulus positions over which the mean Fisher information is calculated for (D). (D) Fisher information (<FI>) increases linearly with population size for all three correlation scenarios over recording sessions. Inset: The boxplot of the Fisher information at a population size of 28 neurons, normalized by the Fisher information at a population size of one neuron for each recording session, shows that the three different correlation scenarios are significantly different (one-way ANOVA: df = 12, p = 3.8 · 10$^{-6}$; SD—I: p = 6 · 10$^{-4}$; SD—SI: p = 6.3 · 10$^{-3}$; I—SI: p = 2.8 · 10$^{-6}$). (E) By transforming the Fisher information to the root Cramér-Rao bound ($\sqrt{CR}$, blue), the standard deviation of the prey location estimate can be compared to the radius of the average prey captured by the fish (0.15 cm, green line). The grey box indicates the range of positions over which averaging is done for (F). (F) The averaged root Cramér-Rao bound (<$\sqrt{CR}$>) vs neural population size decreases to below the average radius of prey at a population of ~19 neurons. Shaded error bars indicate the SEM in (C-F).

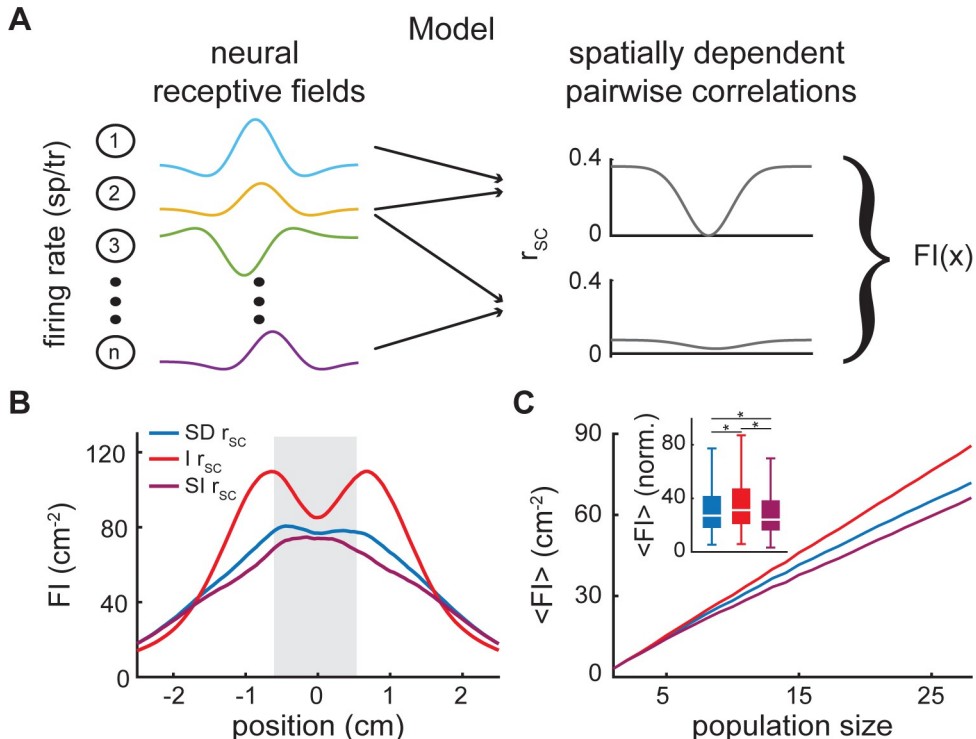

**Fig 7. Modeling receptive field heterogeneities and spatially dependent spike-count correlations.** (A) Schematic of the model in which modeled neural receptive fields (left, colored curves) and spatially dependent pairwise spike-count correlations (right, grey curves) are used to calculate the spatially dependent Fisher information. (B) Fisher information visualized as a function of stimulus position for spatially dependent (SD: blue), independent (I: red) and spatially independent correlation scenarios (SI: purple). (C) Fisher information ($\langle FI \rangle$) as a function of neural population size (averaged between -0.5 and 0.5 cm). Though quite small, the shaded error bars indicate the SEM in (B & C). Inset: Fisher information at the neural population size of 28 neurons, normalized by the Fisher information at a population size of one neuron, is significantly different for the three correlation scenarios: spatially dependent, independent, and spatially independent (Kruskal-Wallis: $p = 8 \cdot 10^{-17}$; SD—I: $p = 3.3 \cdot 10^{-6}$; SD—SI: $p = 6.2 \cdot 10^{-4}$; I—SI: $p = 2.7 \cdot 10^{-17}$). "*" indicates statistical significance.

obtained for spatially independent correlations (compare Figs 7B and 7C to 6C and 6D, respectively; Kruskal-Wallis for 7c: $p = 8 \cdot 10^{-17}$; SD—I: $p = 3.3 \cdot 10^{-6}$; SD—SI: $p = 6.2 \cdot 10^{-4}$; I —SI: $p = 2.7 \cdot 10^{-17}$).

We used the model to investigate the impact of neural heterogeneity on the Fisher information by varying individual parameters (Fig 8). First, we varied the heterogeneity of receptive field position, starting with the receptive fields completely overlapping and gradually increasing the spread so that the receptive fields became more evenly distributed (Fig 8A left and middle panels, respectively). To do so, the standard deviation of the distribution from which the receptive field position is drawn from was increased (see Materials and Methods). Given a fixed population size, the Fisher information increases to a maximum before decaying as the position heterogeneity increases (Fig 8A right panel). Interestingly, the maximum is located close to the level of heterogeneity seen experimentally (black vertical line). This result can be explained by the contribution of individual neurons to the Fisher information as receptive field position is varied (S7A Fig). The rate code assumes that a change in the stimulus will be reflected in a change in firing rate, therefore, a neuron is most informative in the stimulus space where a small change in the stimulus causes a relatively large change in firing rate, in other words where the derivative of the receptive field is largest. For the neurons studied here, the derivative of the receptive field peaks twice on either side of the receptive field center

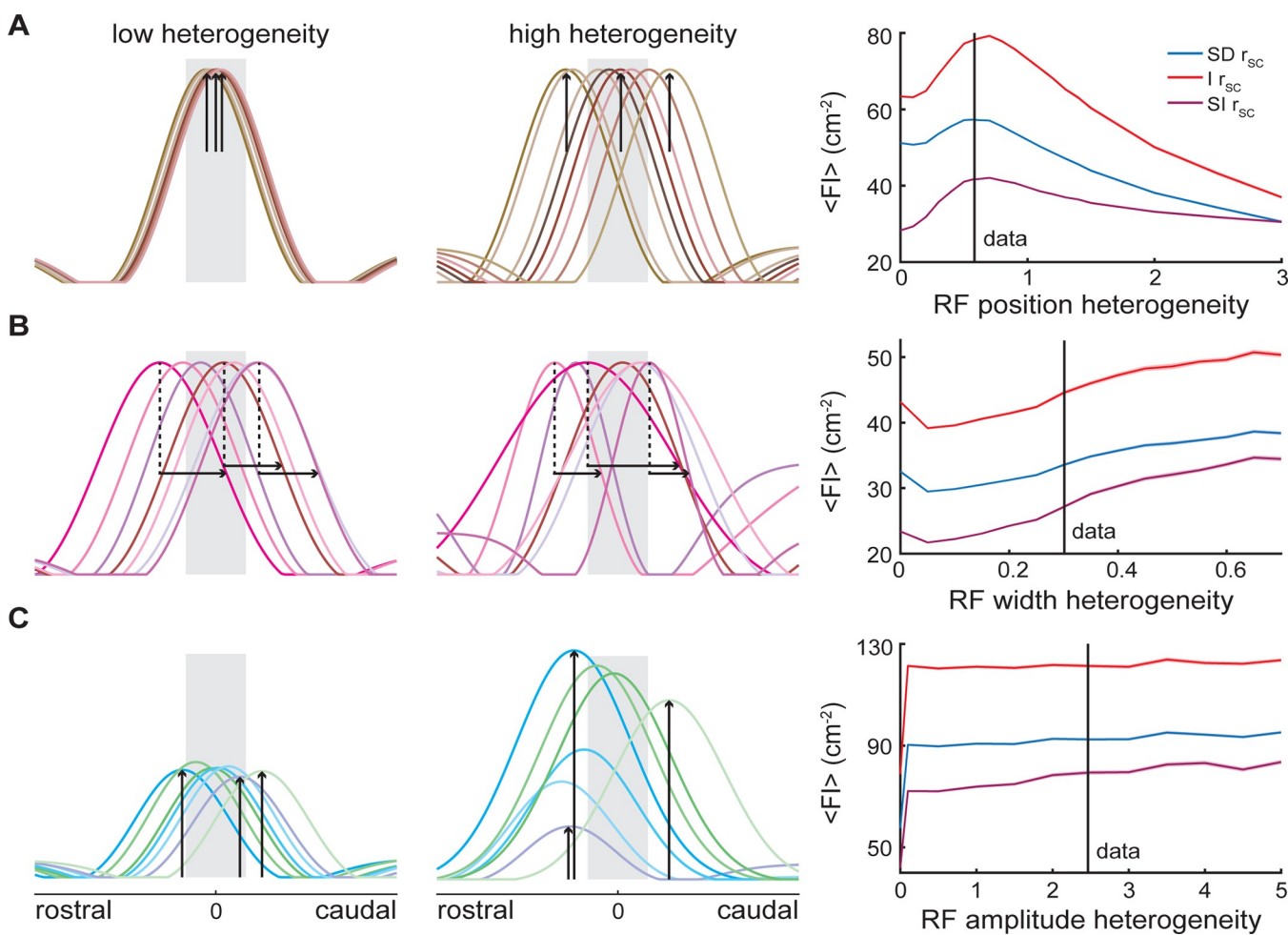

**Fig 8. Receptive field heterogeneities can optimize information transmission.** (A) The level of heterogeneity in terms of the receptive field position was varied from low (left) to high (middle), emphasized by the black arrows. The right panel shows the Fisher information as a function of receptive field position heterogeneity with (blue) and without (red) spatially dependent spike-count correlations, as well as with spatially independent spike-count correlations (purple). Heterogeneity was quantified as the standard deviation of the distribution from which the receptive field position for each neuron is drawn. Though small, the shaded error bars indicate the SEM. In all three correlation scenarios, the Fisher information clearly goes through a maximum as the level of receptive field position heterogeneity is increased. (B) Same as (A) but varying the level of receptive field width heterogeneity. In this case the Fisher information increases in a monotonic fashion with increasing receptive field width heterogeneity. (C) Same as (A) but varying the level of receptive field amplitude heterogeneity. In this case the Fisher information was largely independent of the level of receptive field amplitude heterogeneity.

position. This means that the Fisher information of a single neuron calculated as a function of the receptive field position parameter will peak twice when the steepest portion of either the rostral or caudal side of the receptive field aligns with the 0 cm position. Therefore, the optimal receptive field position parameter space includes a spread of receptive fields that takes advantage of this property.

Next, we fixed the position heterogeneity at the optimal value and kept the amplitude constant but varied the heterogeneity of the receptive field widths by increasing the standard deviation of the distribution from which they are drawn (see Materials and Methods). Examples of populations with low and high width heterogeneity can be seen in the left and middle panels of Fig 8B. There is a gradual trend of increased Fisher information with increased width heterogeneity. This trend can be explained by the differential contributions of narrow receptive fields vs wide receptive fields to Fisher information. The Fisher information increases when the derivative of the receptive field is larger, therefore, given a fixed amplitude, the derivative of a

narrow receptive field is larger than that of a wide receptive field. This relationship turns out to be nonlinear as can be seen for single neurons as width is systematically varied (S7B Fig); with narrow receptive fields contributing significantly more to object localization than wider receptive fields. Therefore, as the population is composed of progressively narrower and wider receptive fields, the increase in Fisher information due to the narrow receptive fields is larger than the loss of information due to the addition of wider receptive fields.

Finally, we fixed the width for the receptive fields and used the position heterogeneity at the optimal value to investigate the change in Fisher information as a function of amplitude heterogeneity which was varied by changing the standard deviation of the distribution from which the receptive field position is drawn (see Materials and Methods). Increasing the heterogeneity of the amplitudes in the population does not change the Fisher information (Fig 8C). This is because the relationship between amplitude and Fisher information for single neurons increases linearly (S7C Fig), meaning that if low and high amplitude receptive fields are added to the population symmetrically, the Fisher information remains flat with increased heterogeneity.

## Discussion

### Summary of results

We investigated how populations of electrosensory neurons in the hindbrain encode prey location. To do this, we recorded the activities of multiple ELL neurons simultaneously in response to a local prey-mimic stimulus located at different positions on the animal's rostro-caudal axis. After having mapped the spatial tuning curves (i.e., receptive fields) of these populations, we analyzed spike train covariations and found strong spatial dependence. Specifically, covariation magnitude was weakest within the overlap middle and strongest at the overlap edges. Further analysis revealed that the spatial dependence of covariation could be explained by changes in correlations between neural activities, as opposed to changes in their variances. Importantly, we found that the spatial dependence of correlations helped offset the overall redundancy that would have been introduced had they been spatially independent, such that accurate information as to prey localisation was available from as few as 19 neurons. Finally, to better understand how different features of spatial tuning curves affected information transmission, we built a mathematical model that incorporated essential aspects of the data. Importantly, our model predicts that heterogeneities in receptive field position seen experimentally play an important functional role towards optimizing information transmission about prey location by ELL pyramidal cell populations. Overall, our results show that accurate information about prey location is contained within the spiking activities of small ELL pyramidal cell populations, which is likely due to spatially dependent covariation as well as heterogeneities in spatial tuning.

### Mechanisms mediating response heterogeneities in ELL pyramidal cell populations

Overall, our results show that ELL pyramidal cell populations display spatially dependent covariance between their activities that help mitigate deleterious effects of redundancy. Additionally, our model predicts that the level of heterogeneity seen experimentally in the spatial tuning functions optimizes information transmission. What is the nature of the mechanisms that mediate heterogeneities in spatial tuning as well as spatially dependent covariation?

It is well known that ELL pyramidal cells display large heterogeneities in their receptive fields [39, 40]. Such heterogeneities are in part due to anatomical differences [55]. For example,

deep ON-type pyramidal cells receive far less feedforward inhibition from local interneurons than their superficial counterparts (see [41] for review), which is thought to underlie the observation that they lack a surround component [39]. Other sources of heterogeneity likely originate from feedback inputs that dominate over feedforward. Indeed, previous studies have shown that ELL pyramidal cells receive large amounts of feedback [56, 57]. One source of feedback, termed indirect, which is diffuse in nature and is activated primarily by global stimulation [58, 59], is unlikely to mediate responses to the spatially localized stimuli considered here. However, another source of feedback, termed direct, which is topographic in nature [53], is more likely to be activated by the stimuli considered here. It has been proposed that the direct feedback pathway has the required properties to act like a sensory searchlight thereby enhancing the sensitivity of the neurons to prey stimuli [53, 60]. A comparison of the predicted spatial extent of the receptive field center of ELL pyramidal cells based solely on feedforward projections [41] versus those measured *in vivo* [39] consistently gave rise to larger estimates *in vivo*, most likely due to the direct feedback pathway. Recent studies have determined novel functions for the direct feedback including synthesizing neural responses to moving objects [61] as well as to envelope stimuli [62]. While these studies were conducted using stimuli that differ from those considered in the current study, it is likely that the direct feedback pathway influences receptive field heterogeneities. Further studies are needed to test this prediction.

In the case of spatially dependent correlations, recent studies have shown that correlations between the trial-to-trial variabilities of neural responses can arise due to feedforward, feedback, and collateral connectivity [32, 63, 64]. In particular, the balance of excitatory and inhibitory input can affect correlation magnitude [54, 65, 66]. Based on the arguments above, it is conceivable that the feedback pathway mediates the spatial dependence of correlations observed here. However, a more likely possibility is that the spatial dependence of correlations is mediated by feedforward input. The fact that previous studies have found that correlation magnitude in ELL pyramidal cells increases with receptive field overlap [29], suggesting that they are primarily due to shared input from electroreceptor afferents that do not display correlations between their trial-to-trial variabilities [67], supports this hypothesis. An interesting possibility is that the decrease in correlation seen in the receptive field center could occur due to "correlation transfer", a mechanism by which increases in signal correlations due to increased stimulus amplitude are accompanied by decreases in noise correlations [68]. This is plausible since the weakest correlations were seen in the center of the pyramidal cell receptive fields, where the sensitivity and, thus, signal correlations should be greatest in magnitude. Further studies are however needed to investigate the contributions of feedforward and feedback processes towards mediating the spatially dependent correlations observed here.

## Functional role of spike-count correlations on population coding of location

Our results show that, while the spike-count correlations between ELL pyramidal cells lead to redundancy overall and thus decrease the amount of information available about location, their spatial dependence helps reduce this redundancy. As such, an important question is, why have spike-count correlations at all? One possibility is that such correlations are an unavoidable consequence of connections between neurons in the brain but are otherwise unimportant. This is because it is possible to construct downstream decoders that ignore the effects of correlations [63, 69–72]. However, the fact that "detrimental noise correlations" have been observed ubiquitously across systems and species [26] and can actually be enhanced during a perceptual task [73] suggests that they serve a beneficial function rather than solely limiting information transmission through increased redundancy. Indeed, while it has been known for some time

that correlations can increase signal propagation [74], recent evidence shows that correlations can enhance the behavioral readout of population activity [75]. Moreover, a recent modeling study predicts that such noise correlations are beneficial by increasing the learning rate and robustness [76]. As such, we propose that the spatially dependent spike-count correlations seen here serve important functions towards adapting to changes in the statistics of natural stimuli. While such adaptation has been observed experimentally [77], this study was conducted using single-unit recordings and, as such, could not assess spike-count correlations. Further investigation is needed to test whether spike-count correlations mediate adaptation and learning.

Our conclusion that the spatial dependence of correlations amongst ELL pyramidal cells can help mitigate their otherwise detrimental effects furthermore agree with theoretical and experimental evidence showing increased information transmission with stimulus dependent correlations [4, 78–81]. It is however necessary to note important differences between these and our current study. For example, one study showed that neurons with high degrees of tuning curve overlap benefit from higher magnitude correlations near the preferred stimulus to reduce redundancy [78]. While other studies, both experimental and theoretical, focus on the sign of the correlated noise, such that correlations are beneficial when they shape the noise in a direction that is orthogonal to that in which the signal varies [3, 4]. The current study adds to the literature in two interesting and possibly related ways. The first is that we focused on a population of neurons whose tuning functions consisted of an antagonistic center-surround organization, as modeled by a difference of Gaussians, whereas most of the studies mentioned above have focused on neurons with bell-shaped tuning curves. The second is that we show that a decrease in the correlation magnitude near the middle of receptive field overlap helps mitigate the otherwise deleterious effects by reducing redundancy, which agrees qualitatively with behavioral studies showing that reduced correlations lead to increased behavioral performance [27, 82–84].

## Functional role of neural heterogeneities in population coding of location

Our results show that there is an optimal value for receptive field position heterogeneity but not for receptive field amplitude or width. As mentioned above, theoretical studies have shown that heterogeneities in neural populations are beneficial for information transmission [1, 15–20, 22, 23]. It is however important to note that previous studies investigating the effects of heterogeneities on coding of spatially varying stimulus attributes such as location have assumed bell-shaped tuning functions (e.g., Gaussian, von Mises, etc. . .) [4, 24, 25]. In contrast, we have considered spatially realistic tuning functions consisting of antagonistic center-surround organization. Our results suggest that the addition of a surround can qualitatively alter the effects of tuning heterogeneities. For example, it was found previously that increasing heterogeneities in the amplitude of the tuning function increased information transmission [24], whereas we found no effect (Fig 8C). These differences are expected as previous studies have shown that Fisher information can display qualitatively different dependencies on population size depending on the assumptions made [21, 25, 85, 86]. Further studies are needed to test the modeling prediction that the effect of tuning function heterogeneities on information will highly depend on their nature as well as the form of the tuning function.

## Implications for other systems

Previous studies have highlighted important anatomical and physiological similarities between the electrosensory and other systems. Specifically, ELL pyramidal cells display antagonistic center-surround organization that is similar to that seen for neurons in early visual pathways

(see [52] for review), including a non-classical portion of the receptive field that serves to enhance responses to high-frequency stimuli [87] as seen in the visual system [88, 89]. Additionally, recent work has shown remarkable similarities in processing strategies between ELL pyramidal cells and central vestibular neurons in macaque monkeys [90–92]. While previous studies in visual areas have considered the effect of spatially dependent noise correlations [4, 7], these did not consider receptive fields consisting of both center and surround. As such, it is likely that our results will be applicable to other systems.

## Materials and methods

### Ethics statement

All animal care and experimental procedures were approved by McGill University's animal care committee (protocol #5285) and were in accordance with the Canadian Council on Animal Care guidelines.

### Animal care

Specimens of either sex of the wave-type weakly electric fish *Apteronotus leptorhynchus* (N = 5) were exclusively used in this study. Fish were obtained from tropical fish dealers and were housed in tanks in groups of up to 10 with the water temperature and conductivity maintained at appropriate levels (26–29° C and 100–800 µS/cm, respectively) as per published guidelines [93].

### Surgery

The surgical techniques used in this study have been previously described in detail [29, 39]. In brief, the experimental tank (30 × 30 × 10cm) was filled with water from the animal's home tank. The water was heated and oxygenated in a reservoir tank and continuously recirculated. The animal was immobilized with an intramuscular injection of 0.1–0.5 mg of tubocurarine (Sigma Aldrich). Upon cessation of gilling, the animal was positioned in the experimental tank and respirated with a constant flow of water through a tube placed in the mouth at a flow rate of ~10 ml/min. Topical lidocaine ointment (5%; AstraZeneca) was applied to anesthetize the skin over the skull and the skull was partly exposed. A head post was then glued to the anterior part of the skull for stabilization and a small window (~5 mm$^2$) was opened to expose the hindbrain over the ELL for electrophysiology. Saline solution was regularly applied to the exposed brain to avoid tissue dehydration.

### Stimulation

The EOD of *A. leptorhynchus* is neurogenic, thus the animal continues to emit its EOD after immobilization with tubocurarine. To record the EOD, electrodes were positioned at the rostral and caudal ends of the fish. Using a function generator (33220 A LXI arbitrary waveform generator, Agilent, Santa Clara, CA USA), a sinusoidal waveform was triggered when the EOD signal crossed zero from below (121 Window discriminator, World Precision Instruments WPI, Sarasota, FL USA), with a frequency approximately 30 Hz higher than that of the EOD to remain synchronized with the EOD. The desired amplitude modulation was generated by multiplying the stimulus waveform with the EOD-triggered waveform (MT3 analog Multiplier, Tucker-Davis Technologies, Alachua, FL USA). The stimulus was isolated from ground (A395 Linear Stimulus Isolator, World Precision Instruments) and delivered to the fish through electrodes. Two steel wire electrodes were positioned approximately 15 cm lateral to either side of the animal and used to deliver spatially extensive (global) stimuli. A dipole was

built from two insulated stainless steel wires with tip spacing ~2 mm and was positioned 1–2 mm lateral from the fish's skin perpendicular to the fish's rostro-caudal axis on the horizontal plane and used to deliver the spatially localized stimuli, as done previously [29, 39].

To study prey localization, we used an amplitude modulation mimicking the timecourse of that caused by prey which consisted of a 4 Hz sinusoidal waveform delivered locally with 15–25% contrast to activate electroreceptors imbedded in the skin within a limited area. The stimulus frequency was chosen because its frequency corresponds to the timecourse of the electric image at a given location on the skin experienced when a foraging fish scans past prey [42]. Prior to beginning the stimulus protocol, the approximate center of the receptive fields of the neurons being recorded was assessed, the dipole was then positioned at the dorso-ventral location that gave rise to the strongest responses and near the rostral edge of the receptive fields, after which the stimulus protocol was started. The stimulus was repeated for 200 trials, with a trial defined as the duration of one full cycle (0.25 s). After completion of each 200-trial stimulus, the dipole was repositioned caudally at 0.5 cm increments and the stimulus was repeated. We collected data at a total of 10 to 15 stimulus positions depending on the recording session such that the majority of the centers and surrounds of the recorded neurons were captured along the rostro-caudal axis. We focused on the rostro-caudal axis because this is the primary axis of motion during the prey detection-to-capture behavior sequence and because the receptive fields are elongated in the rostro-caudal axis compared to the dorso-ventral axis [39, 42, 94]. A zero-mean global Gaussian white noise stimulus that was low-pass filtered (8th order Butterworth, 120 Hz cut-off frequency) was also played globally to characterize neuron type as described below.

### Recordings

Extracellular recordings were collected from ELL pyramidal neurons with Neuropixels probes (Imec, Leuven, Belgium), which allow for simultaneous recording of neural populations [95, 96]. The probe was angled in the ELL to record from pyramidal neurons that are spatially tuned to the same region of the skin, guided by anatomical landmarks and physiological responses. Using spikeGLX (Janelia Research Campus, Howard Hughes Medical Institute), the recordings were digitized at 30 kHz and stored on a hard drive for offline analysis. An automatic spike sorting algorithm followed by manual curation was performed to sort spikes, identify single units, and extract spike times from the recordings. The spike sorting algorithm Kilosort2 (https://github.com/MouseLand/Kilosort2), developed for electrophysiological data with high-channel counts, was used to initially identify and sort single neuron spiking activity. Manual curation of the output of Kilosort2 was done using Phy2 (https://github.com/cortex-lab/phy), a graphical interface for visualization and manual curation of multielectrode data. Well-isolated ELL pyramidal cells, that were stable across the recording session, were identified using firing rate, inter-spike-interval, spike waveform, and autocorrelograms, with clusters merged or split as needed. The sorted neural response activity was then imported into MATLAB (MathWorks Inc., Natick, MA USA) where custom code was used to analyse the data as described below. A total of N = 158 neurons were analyzed over 5 recording sessions (N = 31, 34, 30, 34, 29, respectively). The neuron numbers for each recording session were similar to those obtained in previous studies using Neuropixels probes in the electrosensory system [96, 97].

### Data analysis

Spike times for each neuron were imported into MATLAB and converted to a "binary" time-series (1 when a spike occurs, and 0 otherwise) with a sampling rate of 2 kHz. Note that,

because the inverse of the sampling rate is less than the refractory period of ELL pyramidal cells [98], at most one spike can occur during the corresponding time interval.

## Baseline activity

A recording of 100 s of neural activity in the absence of stimulation was used to calculate the baseline firing rate of each neuron by filtering the binary time-series (2nd order Butterworth filter with a 0.01 Hz cut-off) and averaging the filtered firing rate. Pairwise baseline correlations ($r_{BL}$) were calculated as the Pearson correlation coefficient between all possible pairs at timescales ranging from 2 ms to 2 s. The baseline correlations were similar to those found in previous studies [54]. The baseline correlations (timescale = 0.125 s, to match the prey-mimic stimulus, see below) where then visualized as a function of relative distance between neuron pairs, where the relative distance is the Euclidean distance between the electrode sites on the Neuropixels probe with the largest amplitude spikes for each neuron in a given pair [95, 96]. For pairs composed of the same type of neurons (ON-ON or OFF-OFF, see Noise stimulus section below) the data decayed as a function of distance and was fit with the exponential function $r_{BL}(x) = ae^{bx} + c$, using the MATLAB function "fitnlm" (best-fit parameters a = 0.3722, b = -0.0109 and c = 0.0407). For pairs containing opposite types of neurons (ON-OFF), the data approached zero from the negative side, again showing an exponential relationship with relative distance (best-fit parameters: a = -0.3331, b = -0.0109 and c = -0.0302). In the case of opposite-type pairs it is important to note that the receptive fields (see Prey-mimic stimulus section below) are also opposite because ON and OFF type pairs fire out of phase of each other, therefore the relationship between the correlations and receptive fields are qualitatively similar for all pair types. Due to this conserved relationship across pair types, we visualized the magnitude of the baseline correlations pooled over same and opposite type pairs and again fitted with an exponential curve (i.e., $|r_{BL}(x)| = ae^{bx} + c$; best-fit parameters: a = -0.3360, b = -0.0112 and c = 0.0463).

## Noise stimulus

The neural responses to the 0–120 Hz Gaussian noise stimulus were used to determine if a neuron was ON- or OFF-type by means of a spike triggered average (STA) stimulus analysis, as done previously [99]. Briefly, the stimulus preceding each spike was averaged over all spikes within a time window starting 12 ms and finishing 4 ms before the spike. If the slope of the STA stimulus was positive within this time window, indicating that the neuron responded on average to the increasing phase of the stimulus, the neuron was classified as ON-type. If, in contrast, the slope of the STA stimulus within the same time window was negative because the neuron responded preferentially to the decreasing phase of the stimulus, the neuron was classified as OFF-type. This classification scheme was used to further confirm classification obtained using the receptive field described below.

## Prey-mimic stimulus

In this study, we focused on the firing rate-based receptive field, which was calculated as the sum of spikes over the positive half-cycle of the sinusoidal stimulus (0.125 s) and averaged over trials as a function of stimulus location (see *Stimulation* section for details). Previously, research has shown that these neurons have receptive fields with antagonistic center-surround organization [39, 40], with ON-type neurons increasing their firing rate above baseline in the center and decreasing their firing rate below baseline in the surround. OFF-type neurons show the reverse response, with decreased firing rate in the center and increased firing rate in the surround relative to baseline. For visualizing the population activity across all recorded

neurons, the normalized change in firing rate (Δ firing rate) was calculated by subtracting the baseline firing rate and then normalizing by the maximum firing rate. In order to align the results across experiments, an ON-type population receptive field and an OFF-type population receptive field were calculated by summing all ON-type responses and OFF-type responses respectively and the center of weight of the two was used as the center position (0 cm).

Next, covariance matrices were calculated for each stimulus location. To understand the spatial modulation of the covariance matrices, the pairwise spike-count correlations ($r_{SC}$) were calculated as a function of stimulus location. Briefly, the mean firing rate of a neuron at a given location was subtracted from each trial leaving the residual spike count series. The Pearson correlation coefficient was then calculated on these residuals for each possible pairwise neural combination. Note that because the population is composed of two different pyramidal cell types, ON- and OFF-type, there are two possible pairwise combinations: same-type pairs (ON-ON and OFF-OFF) and opposite type pairs (ON-OFF). Due to our sampling method, which allowed us to collect data from neurons with high degrees of receptive field overlap, for most stimulus positions same-type pairs are positively correlated whereas opposite-type pairs are negatively correlated. The spatial dependence of the correlations is mirrored around zero for these two groups, therefore we used the magnitude of the correlations and pooled the same- and opposite-type pairs. For pairs with high baseline correlation magnitudes, the spike-count correlations were spatially modulated, dipping from baseline values at the edges of receptive field overlap towards zero in regions where the centers of both receptive fields overlap. To characterize this spatial dependence, a baseline correlation threshold of $|r_{BL}| = 0.15$ was used to group the pairs into high and low $|r_{BL}|$ pairs. The depth of modulation of the spike-count correlations was calculated for these two groups as the difference between the maximum and minimum magnitude for each pair, pooled over recording sessions. Finally, the regions of pairwise receptive field overlap were defined as center-center, center-surround, and surround-surround overlap. To test if the spatial modulation of the spike-count correlations was significant for high and low $|r_{BL}|$ pairs, the spike-count correlations were pooled based on the receptive field overlap region and the distributions were tested for significant differences.

## Coding

To quantify the precision with which a population of neurons can estimate the location of prey, the linear Fisher information was calculated [4, 8, 85]. The Fisher information (FI) is a quantification of the inverse variance of the optimal estimator of a variable and is calculated as a function of the slope of the receptive fields and covariance matrices as follows:

$$FI(x) = RF'(x)^T \Sigma^{-1}(x) RF'(x),$$

where x is the stimulus position, $RF'(x) = [f_1'(x), \ldots, f_n'(x)]$ is the matrix whose elements are the slope of the receptive fields, $f_i(x)$, with $i = 1, \ldots, n$, for n neurons in a given recording session with the superscript T indicating the transpose operation, and $\Sigma^{-1}(x)$ is the inverse covariance matrix at position x. Each neuron's receptive field was fit with a smoothing spline function and then the slope of each receptive field fit was calculated using the MATLAB function "differentiate". The slope of the receptive fields and covariance matrices were calculated as the average of these same variables between two locations in order to reduce the variability due to a finite number of trials as done previously [100].

In order to investigate the effect of the spatially dependent spike-count correlations on the Fisher information, two alternatives to this quantification were performed: an independent scenario and a spatially independent scenario. For the former, we rendered the neurons independent from one another within a given population by setting all of the off-diagonal values of

the covariance matrices to zero and recalculating the Fisher information as done previously [4]. Randomly shuffling the trials using the function "randperm" in MATLAB prior to computing the covariance matrix and averaging over 20 different permutations led to qualitatively similar results as expected. For the latter, we decomposed the covariances into variances and correlations and replaced the stimulus dependent correlations with the baseline correlation (timescale = 0.125 s) for all stimulus positions. The covariances were then recalculated using these values, thereby building spatially independent correlations. Note that in both scenarios, the Fisher information calculation still includes the spatially dependent receptive fields and variances.

We also considered how the Fisher information grows as a function of population size for all correlation scenarios (i.e., spatially dependent, independent, and spatially independent). To do so, we averaged the Fisher information between locations x = -0.5 and 0.5 cm (<FI>), which was the spatial region to which the majority of the recorded neurons were maximally sensitive. We then subsampled the data by bootstrapping 250 times for neural population sizes of 1 to 28 neurons (limited by the experiment with the fewest simultaneously recorded neurons). To compare the three cases at the maximum population size across experiments, we normalized by the Fisher information at a population size of one neuron for each experiment.

To compare the accuracy of the location estimate with the size of the prey, we converted the Fisher information to the Cramér-Rao bound. The Cramér-Rao bound is greater than or equal to the inverse Fisher information and is the lower bound on the variance of the estimate:

$$CR(x) = var(x_{est}) \geq \frac{1}{FI(x)}.$$

By taking the square root of the Cramér-Rao bound we were able to compare the standard deviation of the estimate of prey location with the radius of the average prey item, ~0.15 cm, found in the digestive tract [42]. Using the same spatial averaging and bootstrapping described above for Fisher information vs population size, we calculated the Cramér-Rao bound vs population size to determine how many neurons are required to estimate the location of a prey item within the size of the prey.

## Model

A mathematical model was built to better understand how specific features of the data (e.g., receptive field size) influence results seen experimentally. As discussed above, the Fisher information calculation requires both the spatial receptive fields and the covariance matrices, which were decomposed into the spatially dependent variances and correlations per the relationship:

$$\rho(x, y) = cov(x, y)/\sqrt{var(x) \cdot var(y)}.$$

The receptive fields of the model neurons were determined by first fitting each recorded neuron across sessions with a difference of Gaussians:

$$RF(x) = \alpha_c e^{.5\left(\frac{x-\mu}{\sigma_c}\right)^2} - \alpha_s e^{.5\left(\frac{x-\mu}{\sigma_s}\right)^2},$$

where $RF(x)$ is the receptive field as a function of position x, μ is the position of the center of both Gaussians, $\alpha_i$ is the amplitude, $\sigma_i$ is the width and subscripts *c* and *s* refer to the center and surround Gaussian respectively. The fitted parameters (position (μ), amplitude (α) and width (σ)) were pooled across all 158 recorded neurons. The resulting parameter distributions were then fitted, and numbers drawn from these randomly to generate model neural receptive fields. Specifically, the distribution of receptive field positions relative to the center position at

0 cm was fitted with a normal distribution (μ = 0, σ = 0.58). The position of each model receptive field was drawn from this distribution. Next, the widths of the center Gaussians were drawn from a gamma distribution (α = 2.28, β = 0.28) fitted to center widths from the data. The surround widths were considered relative to the center widths because they are necessarily larger: when the center widths are plotted against the surround widths all data points fall on or above the unity line. The surround widths were built from the center width plus a number drawn from the gamma distribution (α = 0.61, β = 2.14) fitted to the difference in width between surround and center. Finally, to model the amplitude of the receptive fields, we started by comparing the fitted center and surround amplitudes which show a strong negative correlation (r = -0.9998), with the surround amplitude necessarily smaller in magnitude than the center amplitude. Therefore, amplitudes of the receptive field centers were drawn from a fitted t-distribution (μ = 4.34, σ = 2.48, ν = 0.51). The amplitudes of the surrounds were set equal to the center amplitude plus a value drawn randomly from a gamma distribution (α = 2.28, β = 0.96) fitted to the distribution of differences between center and surround amplitudes. Once the parameter space for a given neuron was determined as explained above, the receptive field was calculated as a function of space. For OFF-type neurons only, the receptive field was multiplied by -1. Finally, because firing rate is necessarily positive, the receptive fields were shifted upward. Some neural receptive fields from the data show a flat section or rectification as the firing rate approaches zero (in the center of OFF-type neurons and in the surrounds of ON-type neurons). Including this rectification improved the model significantly, therefore we incorporated it by adding noise to how much we shifted the receptive field upwards and then setting the minimum firing rate to 0.1 spikes/trial.

As the average number of neurons per recording session was N = 31.6, populations of 32 neurons were modeled (16 ON-type, and 16 OFF-type). The neurons were ordered by their center position and randomly assigned channel numbers used to calculate a relative distance as done for the data. This caused the neuronal pairs with smaller relative distances to show more receptive field overlap than pairs with larger relative distances, allowing there to be a relationship between receptive field overlap and neural distance.

We found experimentally that the variance in neural activity increases with firing rate. To model this dependency, we graphed the variance vs firing rate for all recorded neurons at all locations, binned and averaged the data in increments of 0.2 spikes/trial and fit the results with a linear function ($var(x) = A * firing\ rate(x) + B$). We obtained A = 1.17 and B = 0.24 using a linear least-squares fit. This function was then used to model the variance of each neuron at each position.

The spike-count correlations ($r_{SC}$) were modeled using baseline correlations ($r_{BL}$) and the amount of receptive field overlap. To begin, the baseline correlations were modeled by fitting a sum of exponentials to the $|r_{BL}|$ vs relative distance data. To do so, the data was binned in 10 μm increments and averaged. These averaged data were then fit with a sum of two exponentials to which random normally distributed noise (μ = 0, σ = 0.05) was added. Next, to model the spatial dependence of the $r_{SC}$, the geometric mean of the center Gaussians of each pairwise combination of neurons was calculated and subtracted from 1, such that the curve dips to zero in regions where the centers overlap. For same-type pairs this curve ranges between 0 and 1, and for opposite-type pairs it was flipped around the zero axis to range between -1 and 0. Finally these curves were multiplied by the modeled $r_{BL}$ so that the curve approaches $r_{BL}$ at the edges of receptive field overlap as seen in the data.

The modeled population of spatially dependent receptive fields, variances and correlations were then used to calculate the spatially dependent Fisher information and the Fisher information as a function of population size, as described in the *Data analysis* section. The independent and spatially independent cases were also calculated as described previously. Each

iteration of this process was treated as a single experiment, a total of 1000 experiments were performed to reproduce the results from the data, as well as to investigate the different model parameters described below.

After reproducing the results given the parameter space found in the data, we extended the model to investigate lower and higher degrees of heterogeneity in the position, width, and amplitude parameter spaces for all three correlation cases. To begin, we calculated the Fisher information contained in the activity of a single neuron as we varied each parameter while fixing the other two parameters. We then turned to populations of neurons, again fixing two of the parameters across all neurons, while gradually increasing the heterogeneity of the third to establish the impact of that parameter on information. First, we varied the heterogeneity of spatial position while holding the amplitude and width of the receptive fields constant across all populations. Initially, all receptive fields in the population were centered at 0 cm, such that the receptive fields were completely overlapped, and then the spread of the receptive field positions were gradually increased by increasing the standard deviation of the normal distribution from which the receptive field positions were drawn (standard deviations: 0, 0.1, 0.2, 0.3, 0.4, 0.5, 0.58, 0.6, 0.7, 0.8, 0.9, 1, 1.1, 1.2, 1.3, 1.4, 1.5, 2, 2.5, 3, where the data has an average standard deviation of ~0.58). Next, we held the amplitudes constant, allowed the position to vary per the normal distribution from the data, and gradually increased the standard deviation of the distribution from which the receptive field widths was drawn. In the data the widths were drawn from a gamma distribution, but because it is not symmetric, we replaced it with a normal distribution so that the inclusion of narrower and wider receptive fields occurred symmetrically (standard deviations: 0.05, 0.1, 0.15, 0.2, 0.25, 0.3, 0.35, 0.4, 0.45, 0.5, 0.55, 0.6, 0.65, 0.7, where the data has an average standard deviation of ~0.28). Finally, we held the receptive field widths constant, allowed the position to vary per the normal distribution from the data, and varied the amplitude heterogeneity. Again, because the distribution of amplitudes in the data is not symmetric, we were not able to simply use the same distribution and increase the distribution spread. Instead, we used a normal distribution centered at the center of weight of the original distribution and gradually increased the standard deviation (standard deviations: 0.1, 0.5, 1, 1.5, 2, 2.48, 3, 3.5, 4, 4.5, 5, where the data has an average standard deviation of ~2.48).

## Statistics

Values are reported as mean ± SEM. Unless otherwise indicated, statistical tests were performed using a one-way ANOVA or Kruskal-Wallis, depending on the normality of the data per the Lilliefors test, with Bonferroni test correction. Where indicated, outliers were removed using the generalized extreme Studentized deviate test for outliers in MATLAB (i.e., the "rmoutliers" function) which defines an outlier as a value greater than three scaled median absolute deviations from the median. We note that this constitutes a small percentage of the data (<2.3%) and that removing these outliers does not affect the qualitative nature of our results.

## Supporting information

**S1 Fig. Calculation of spike-count correlations.** (A) The spike train of an example neuron (n1) is shown, with each trial indicated by alternating grey and white bars and the trial number above. The spike counts for each trial are indicated below. (B) The spike counts, or firing rate, of 3 example neurons (n1 left, n2 middle and n3 right) are shown for 30 trials each. The trial-averaged firing rate over all 200 trials in the data is also shown (black horizontal line) for each neuron. (C) The residual spike counts (the spike counts for each trial minus the trial-averaged firing rate) vary around zero. (D) The color plot shows the residuals of one neuron vs another.

The color shows the number of trials in each square. There is a strong positive spike-count correlation for the pair on the left and no significant correlation for the pair on the right (red fitted line: left pair: n1 and n2, rSC = 0.51, p = 1.1 · 10−27; right pair: n2 and n3, rSC = - 0.03, p = 0.54). Showing that for the pair on the left, when one neuron tends to fire above its mean, the other does as well, whereas the pair on the right tend to vary around their respective trial-averaged firing rate independent from each other.
(TIF)

**S2 Fig. Distance dependence of raw spike-count correlations by pair type.** Pairwise baseline correlation (rBL) values approach zero as a function of increasing relative distance between neurons. The baseline correlation values of same-type pairs (ON-ON and OFF-OFF; blue circles) follow an exponentially decreasing trend as a function of distance (blue fitted line) and the opposite-type pairs (ON-OFF; green triangles) show the same exponential trend approaching zero but from the negative direction (green fitted line). In the case of opposite-type pairs it is important to note that the receptive fields are opposite as well because ON and OFF type pairs fire out of phase of each other, therefore the relationship between the correlations and receptive fields are qualitatively similar for all pair types.
(TIF)

**S3 Fig. Model receptive fields.** (A) Example ON-type receptive field (black) is shown with the fitted difference of Gaussians curve (purple). (B) Schematic showing the parameters of the difference of Gaussians fit: the left and middle curves (black) show the center and surround Gaussians and the right curve (black) shows the fitted receptive field. The blue circles mark the center position of the receptive field. The dark green horizontal bracket is the width of the center, the light green bracket is the difference between the widths of the center and surround. The vertical dark orange bracket is the center amplitude, and the light orange bracket is the difference between the center and surround amplitude. (C) Receptive field positions: The distribution of the positions of all neurons pooled across sessions (blue) was fitted with a normal distribution (black). (D) Receptive field widths: For all fitted neurons, the width of the surround Gaussian is larger than the width of the center as the data (left panel; green dots) falls above the unity line (dashed grey line). Therefore, the distribution of the center Gaussian widths was fitted with a Gamma distribution as was the distribution of the difference between the two (surround-center). After drawing randomly from both distributions, the surround was modeled as the sum of the two. (E) Receptive field amplitudes: The surround amplitude vs center amplitude (left panel) demonstrates that the center amplitude is always larger than the surround amplitude (orange dots lie above the dashed grey unity line). The distribution of center amplitudes (middle panel; dark orange) was fitted with a t-distribution and the difference distribution (center–surround) was fitted with a Gamma distribution (right panel; light orange). After drawing randomly from both distributions, the first was used as the center amplitude and the second was subtracted from the first to model the surround amplitude. (For all distribution parameters see Materials and Methods.)
(TIF)

**S4 Fig. Model variances and correlations.** (A) The variances of the neural firing rates in the data increase with the average firing rate. Prior to fitting, the data was binned and averaged (black dots) then fit with a linear function (red line). (B) To fit the |rBL| as a function of relative distance, the data was binned and averaged, and a sum of exponentials was fitted (red curve). (C) To confirm that as the stimulus approaches the edge of the population of receptive fields, the rSC approach rBL, the spike-count correlations at the stimulus positions at the rostral and caudal edges of the recording positions were visualized vs the rBL, which shows a strong linear

relationship (see Materials and Methods for details.)
(TIF)

**S5 Fig. Model correlations.** The spatially dependent correlations are modeled as a function of the overlap of the pair of receptive fields (top row; neuron 1 blue, neuron 2 purple). The geometric mean (3rd row; grey) of the absolute value of the receptive field centers (2nd row) is normalized to range between the baseline correlation assigned to that pair and zero (bottom row; black). Three examples are provided: a high rBL, opposite-type pair (left), a low rBL, opposite-type pair (middle), and a moderate rBL, same-type pair (right).
(TIF)

**S6 Fig. Model covariance matrices.** The model covariances are shown for three different stimulus positions: at the center of the population of receptive fields where there is primarily center-center overlap (center panel) and near the edges of the population of receptive fields where there is surround-surround overlap (right panel) or minimal to no overlap (left panel). These results demonstrate that the covariances calculated from the modeled variances and correlations reproduce trends seen in the data. Inset: the distributions of the covariance magnitudes are significantly different across these three stimulus positions (Kruskal Wallis: $p = 1.5 \cdot 10^{-190}$). "*" indicates statistical significance.
(TIF)

**S7 Fig. The contribution of single neurons to information transmission varies as a function of different model parameters.** The left column shows example receptive fields, and the right column shows the Fisher information ($<FI>$) averaged between -0.5 and 0.5 cm (grey region in left panels). (A) The receptive field position parameter is varied, while holding the receptive field width and amplitude parameters constant. (B) The receptive field width parameter is varied, while hold position and amplitude constant. (C) The receptive field amplitude parameter is varied while hold position and width constant.
(TIF)

## Author Contributions

**Conceptualization:** Myriah Haggard, Maurice J. Chacron.

**Data curation:** Myriah Haggard.

**Formal analysis:** Myriah Haggard.

**Funding acquisition:** Maurice J. Chacron.

**Investigation:** Myriah Haggard.

**Methodology:** Myriah Haggard.

**Project administration:** Maurice J. Chacron.

**Resources:** Maurice J. Chacron.

**Software:** Maurice J. Chacron.

**Supervision:** Maurice J. Chacron.

**Validation:** Myriah Haggard, Maurice J. Chacron.

**Visualization:** Myriah Haggard, Maurice J. Chacron.

**Writing – original draft:** Myriah Haggard, Maurice J. Chacron.

**Writing – review & editing:** Myriah Haggard, Maurice J. Chacron.

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
