## [Decision Letter · Decision Letter 0]

8 Jan 2023

Dear Dr. Chacron,

Thank you very much for submitting your manuscript "Coding of spatial position by heterogeneous neural populations with spatially dependent correlations" for consideration at PLOS Computational Biology. As with all papers reviewed by the journal, your manuscript was reviewed by members of the editorial board and by several independent reviewers. The reviewers appreciated the attention to an important topic. Based on the reviews, we are likely to accept this manuscript for publication, providing that you modify the manuscript according to the review recommendations.

Sincerely,

Jonathan Rubin

Academic Editor

PLOS Computational Biology

Lyle Graham

Section Editor

PLOS Computational Biology

Reviewer's Responses to Questions

**Comments to the Authors:**

Reviewer #1: The authors study the spatial dependence of correlations in the weakly electric fish, and the role of this structure in coding accuracy. They find that correlations reduce accuracy overall, but their spatial dependence increases accuracy as quantified by Fisher Information. They reproduce these effects in a computational model and explore the role of receptive field heterogeneity.

The problem is well motivated and the results are clearly presented. The authors use well known approaches to answer their question in a straightforward manner, which I view as a strength of the paper. The manuscript is suitable for publication in PLoS Computational Biology. I only have a few very minor suggestions that could help improve the paper.

1. Line 178: Specify what measure of covariance is used (e.g., spike count covariance) and point to the Methods.

2. Figure 5A: It would be interesting to also see the raw correlation values (without the absolute value), even if only as a Supplementary Figure. This could be important because negative correlations impact coding differently than positive correlations (see, e.g., Moreno-Bote et al, 2014). If the average sign of correlations differs based on whether neurons are ON-ON, ON-OFF, or OFF-OFF pairs, then these values could be shown as different colored points in the scatter plot.

3. Figure 5A: Since the scatter plot is so noisy. it would be nice to see a curve fit to the scatter plot. This could also be done for the Supplementary Figure suggested above.

Reviewer #2: Summary:

In the present study, Haggard and Chacron analyze the effects of correlations in the activity of neurons from the electric fish Apteronotus leptorhynchus in the coding of the spatial location of a stimulus along the rostro-caudal axis of the animal. The authors report that the neural activity presents spatially structures correlation patterns, and that these patterns allow to resolve the problem poised by information limiting correlations. Furthermore, through mathematical modeling the authors show that heterogeneity in the neural tuning and neural response magnitude help to optimize information transmission.

Strengths:

Overall, this is a well-executed study and a very nice manuscript. One of its main strengths is the use of a “non-standard” experimental model in the neutral coding literature. This allows to study the effects of receptive field heterogeneity in information transmission. While similar work has been done for other sensory systems in mammals, the use of the electro-sensory system in this model allows to control variables that are more difficult to manipulate in other systems/organisms.

Weaknesses:

The size of the studied neural populations is small, compared to the population sizes recorded in similar studies. Also, the manipulations performed to compute 1) independent and 2) spatially independent Fischer information are in a completely theoretical way. Although this is perfectly valid, it would be good to do a data shuffling instead of manually manipulating the covariance matrices. Similar results would be expected from both approaches.

Major comments:

1) Line 228: The authors report that one of the main findings is that the spatial dependence of the covariance is due to the spatial dependency of noise correlations. Why is this surprising? Given that correlation is defined as the normalized covariance, it is only expected that if one of them shows spatial structure, the same will be observed in the other.

2) Lines 264-279: As mentioned above in “Weaknesses”. It is important to have a complimentary analysis in which independence and spatial independence are achieved through data shuffling, rather than direct manipulation of the covariance matrices. This way we would have “independent” information levels based on the data available, rather than the idealized scenario that results from setting off diagonal values to exactly zero by hand.

Minor comments:

3) Line 121: Please indicate what was the threshold here. I know it is in the methods, but there is no reason to bury that number in there.

4) Figure 5 legend (line 253) and Methods (line 790): please explain the criteria used to reject outliers (sig level, k)

5) Figure 6C (line 292): what was the population size for this figure?

6) Figure 8: How exactly was heterogeneity quantified? What does the horizontal axis in the plots indicate?

7) Lines 480-482: It is a matter of debate if correlations have a role, or if they are simply due to the data processing inequality principle. There is certainly evidence that in some cases correlations are actively shaped to enhance specific information, but there are instances in which this is not clear. For the sake of completeness, it would be good to mention both sides.

8) Line 661: a clearer explanation of the procedure to calculate the slope of the receptive field is needed.

9) Line 740: where do the slope and intercept for the linear model come from? For all other parameters it is stated that they are derived from the observed neural activity, one would assume the same applies for this linear model. Better to explicitly indicate the source.

10) Line 789: “normalcy” should be “normality”.

Reviewer #3: This paper contains a masterful analysis of population coding of object location in the electrosensory system of the electric fish.

The conclusions about advantages for coding of heterogeneity of receptive fields and of spatially dependent correlations are well sustained by the data and are of wide interest.

The experiments and analyses are excellent and the paper is clear and well structured. The only suggestions for improvement (listed below) regard textual clarifications of specific aspects and better placing the paper in the context of the existing literature and better individuating its novelty.

One main conclusion is about the advantages for population codes of heterogeneous neural properties. Advantages for heterogeneous properties have been shown before by theoretical studies. Some of these are cited by the authors (Shamir and Sompolinksy 2006) . Some other prominent papers on heterogeneity and population coding are mot cited, for example:

Ecker, A. S., Berens, P., Tolias, A. S. & Bethge, M. The effect of noise correlations in populations of diversely tuned neurons. J. Neurosci. 31, 14272–14283 (2011).

Wilke, S. D. & Eurich, C. W. Representational accuracy of stochastic neural populations. Neural Comput. 14, 155–189 (2002).

It would be useful to cite also these papers and to better discuss the specific new insights of the present paper with respect to the older theories of heterogeneity vs population codes.

The second main conclusion is about the location dependence of spike count correlations and its effect on decreasing the information-limiting effects of positive signal-noise correlations (“mitigating redundancy”). A location dependence of noise correlations which increases the population information has been reported (both among neurons, among astrocytes, and between neurons and astrocytes) in the hippocampus of mice during spatial navigation (Curreli et al PloS Biology 2022). More generally, beyond location coding, the problem of how modulations of correlations influence positively population codes has been the subject of much theoretical work, very relevant to the results presented here (e.g. Pola et al Network 2003, , Josic, Shea- Brown, Doiron, de la Rocha,

Neural Comput. 2009, Azeredo da Silveira and Rieke Annu. Rev. Neurosci. 2021 and cited references). It would be interesting if the authors cite other evidence of information-enhancing location modulations of correlations and their contribution to coding (including the one suggested above) and discuss more differences, similarities, relationships and implications of their results with respect to previous works reporting related findings or theoretical considerations on stimulus-dependent correlations.

On page 29 (Discussion), the authors speculate that noise correlations may be good for perceptual tasks even if they decrease information. Recent evidence to support this was presented in Valente et al Nature Neurosci 2021, which should be mentioned. Also, it should be mentioned that correlations may not only increase learning rate and robustness but also signal propagation (as shown the 1900s studies of Christof Koch, Emilio Salinas etc).

Introduction and abstract place a lot of emphasis on the role of population codes not only for information encoding but also for generating behaviors, which is a nice question. However, in my understanding, the paper results are mor focused on information coding than in the subsequent generation of behaviors. It may be worth reconsidering the balance of the opening sentence of the abstract and introduction.

The title is nice but al little too generic. I feel that having in the title of the paper object locations and electric fish would help readers.

**Have the authors made all data and (if applicable) computational code underlying the findings in their manuscript fully available?**

Reviewer #1: Yes

Reviewer #2: Yes

Reviewer #3: Yes

PLOS authors have the option to publish the peer review history of their article (what does this mean?). If published, this will include your full peer review and any attached files.

Reviewer #1: No

Reviewer #2: No

Reviewer #3: No

Figure Files:

Data Requirements:

Reproducibility:

References:

---

## [Decision Letter · Decision Letter 1]

9 Feb 2023

Dear Dr. Chacron,

We are pleased to inform you that your manuscript 'Coding of object location by heterogeneous neural populations with spatially dependent correlations in weakly electric fish' has been provisionally accepted for publication in PLOS Computational Biology.

Best regards,

Jonathan Rubin

Academic Editor

PLOS Computational Biology

Lyle Graham

Section Editor

PLOS Computational Biology

Reviewer's Responses to Questions

**Comments to the Authors:**

Reviewer #2: The authors have done a good job addressing my comments. The manuscript was very strong on the initial submission, thus my comments and those of the other reviewers were minor.

I have no further comments or questions.

Reviewer #3: I am satisfied with the revisions.The authors used well the suggestions of all reviewers to improve the paper.

**Have the authors made all data and (if applicable) computational code underlying the findings in their manuscript fully available?**

Reviewer #2: Yes

Reviewer #3: Yes

PLOS authors have the option to publish the peer review history of their article (what does this mean?). If published, this will include your full peer review and any attached files.

Reviewer #2: No

Reviewer #3: **Yes: **Stefano Panzeri

---

## [Editor Report · Acceptance letter]

28 Feb 2023

PCOMPBIOL-D-22-01507R1 

Coding of object location by heterogeneous neural populations with spatially dependent correlations in weakly electric fish

Dear Dr Chacron,

I am pleased to inform you that your manuscript has been formally accepted for publication in PLOS Computational Biology. Your manuscript is now with our production department and you will be notified of the publication date in due course.

With kind regards,

Anita Estes
